# Which Gelatin and Antibiotic Should Be Chosen to Seal a Woven Vascular Graft?

**DOI:** 10.3390/ijms25020965

**Published:** 2024-01-12

**Authors:** Irina Yu. Zhuravleva, Aldar A. Shadanov, Maria A. Surovtseva, Andrey A. Vaver, Larisa M. Samoylova, Sergey V. Vladimirov, Tatiana P. Timchenko, Irina I. Kim, Olga V. Poveshchenko

**Affiliations:** 1E. Meshalkin National Medical Research Center of the RF Ministry of Health, 15 Rechkunovskaya St., Novosibirsk 630055, Russia; a_shadanov@meshalkin.ru (A.A.S.); mfelde@ngs.ru (M.A.S.); vaver_a@meshalkin.ru (A.A.V.); l_samoylova@meshalkin.ru (L.M.S.); vladimirov_s@meshalkin.ru (S.V.V.); t_timchenko@meshalkin.ru (T.P.T.); kii5@yandex.ru (I.I.K.); poveschenkoov@yandex.ru (O.V.P.); 2Research Institute of Clinical and Experimental Lymphology, Branch of the Federal Research Center «Institute of Cytology and Genetics SB RAS», 2 Timakova St., Novosibirsk 630060, Russia

**Keywords:** woven vascular graft, graft sealing, type A and type B gelatins, gelatin film with antibiotic

## Abstract

Among the vascular prostheses used for aortic replacement, 95% are woven or knitted grafts from polyester fibers. Such grafts require sealing, for which gelatin (Gel) is most often used. Sometimes antibiotics are added to the sealant. We used gelatin type A (GelA) or type B (GelB), containing one of the three antibiotics (Rifampicin, Ceftriaxone, or Vancomycin) in the sealant films. Our goal was to study the effect of these combinations on the mechanical and antibacterial properties and the cytocompatibility of the grafts. The mechanical characteristics were evaluated using water permeability and kinking radius. Antibacterial properties were studied using the disk diffusion method. Cytocompatibility with EA.hy926 endothelial cells was assessed via indirect cytotoxicity, cell adhesion, and viability upon direct contact with the samples (3, 7, and 14 days). Scanning electron microscopy (SEM) and energy dispersive spectrometry (EDS) were used to visualize the cells in the deep layers of the graft wall. “GelA + Vancomycin” and “GelB + vancomycin” grafts showed similar good mechanical characteristics (permeability~10 mL/min/cm^2^, kinking radius 21 mm) and antibacterial properties (inhibition zones for *Staphilococcus aureus*~15 mm, for *Enterococcus faecalis*~12 mm). The other samples did not exhibit any antibacterial properties. The cytocompatibility was good in all the tested groups, but endothelial cells exhibited the ability to self-organize capillary-like structures only when interacting with the “GelB + antibiotics” coatings. Based on the results obtained, we consider “GelB + vancomycin” sealant to be the most promising.

## 1. Introduction

Thoracic aorta aneurysm is diagnosed in 6–10 cases per 100 thousand population [1,2]. In half of the cases, these are patients with acute or subacute aortic dissection, requiring emergency surgery due to the extremely high mortality rate with conservative treatment, which is 95–96% [3,4].

The total reconstruction of the thoracic aorta is a very complex surgical procedure; it rapidly advanced after the development of the “elephant trunk” (ET) technique by Hans Borst in 1983 [5]. The progress has further accelerated since 2003 when the “frozen elephant trunk” (FET) was invented [6].

The FET technique is a hybrid operation, which is a combination of a minimally invasive, sutureless implantation of a stent graft into the descending aorta with traditional open surgery on the arch and ascending aorta. It has gained popularity because of surgical procedure alleviation for both the doctor and the patient [7]. The hybrid procedure requires a hybrid graft including two or more parts. The mandatory part is a stent graft of the descending thoracic aorta connected to a non-stented aortic arch prosthesis. The hybrid prosthesis can be supplemented with branches (from 1 to 3) for connection with the brachiocephalic arteries if they have undergone dissection. Over the past 20 years, approximately 1000 publications have been found in PubMed regarding the techniques, devices, and clinical outcomes of hybrid thoracic aortic surgery using FET.

The designs of hybrid prostheses continue to be improved. However, in addition to their design features, a number of problems with graft materials and sealing techniques remain unresolved.

Today, 95% of all hybrid prostheses in the world market are made of polyester [8], which can be woven and knitted. There are three main patterns used in the manufacturing of woven tack: plain, twill, or satin. For knitted grafts, weft and warp knitting designs are used [8,9,10]. Plain woven fabric is the most popular because it is denser and stronger, less permeable to blood, and less likely to unravel during intraoperative modeling and anastomosis. However, even when using the same plain weaving method, the overall pattern and permeability of the fabric depend on the configuration and density of the fibers in the thread, as well as on the weaving equipment and technique [11]. Knitted prostheses are softer, more flexible, and stretchable compared to woven ones, which, on the one hand, is an advantage, but on the other, often leads to dilatation and aneurysms of the prostheses [8,12].

For woven and knitted aortic prostheses, various compositions and methods of sealing have been developed, including those containing antibacterial drugs. Gelatin-based (Gel) sealants are most popular due to their ability to form low-viscosity hydrogels [13,14], which form a thin elastic film on the inner and outer surfaces of a porous prosthesis after drying [15,16]. Gelatin sealing techniques vary widely, but they all require that the coating be elastic, uniform in thickness, firmly bonded to the surface, not peeled off during storage and intraoperative manipulation, and also do not significantly increase the rigidity of the graft. In this regard, the differences between sealing with gelatins A (GelA) or B (GelB) are interesting since they differ in the structure and properties of the gels they form.

In addition, the choice of the optimal antimicrobial agent remains a very pressing problem. Bacterial invasion with the subsequent development of generalized infection is one of the most severe complications accompanied by high mortality. The frequency of this complication in the early postoperative period is 1–6%, and Gram-positive bacteria (*S. aureus.* and *S. epidermidis*) are the main infectious agents in 60–70% of cases [17]. In this regard, the antimicrobial additives to the sealant are necessary to prevent early contamination (until the protein layers on the inner surface of the graft are absorbed).

W. S. Moore et al. were the first to apply antibiotics in 1981 to seal vascular grafts [18]. Antibiotics can be applied either by soaking the graft in an antibiotic solution, or by introducing the drug into the composition of the sealing. Both methods provide immediate antibacterial protection of the implant; however, with soaking (passive adsorption), the antibiotic depot depletes very quickly [19]. If the antibiotic is bound to the main component of the sealant, it is released into the bloodstream gradually [20,21,22]. In this case, the release rate depends on the type of antibiotic and the composition of the coating. Thus, J. Galdbart et al. showed that albumin or gelatin sealants retained antibiotics for a longer time compared to collagen [21]. When comparing the six antibiotics, the authors found that rifampicin was the slowest to be released from these coatings.

Rifampicin has been the most often used drug for the impregnation of vascular grafts for many years [23]. However, it has now been proven that this drug has a number of disadvantages. First, it promotes the development of antibiotic resistance in microorganisms and is also ineffective against methicillin-resistant *Staphylococcus aureus* [24], which, in turn, is associated with extremely poor clinical outcomes. Second, rifampicin is highly cytotoxic [25,26]. There is an opinion that rifampicin inhibits all types of cells (endothelial, smooth muscle, and fibroblasts) that come into contact with the implant [26].

All antibiotics are expected to be cytotoxic because this is their main effect on microbial cells. It should be noted that there are a few studies on the effectiveness and cytotoxicity of antibiotics included in gelatin sealing [23], and rifampicin appears more often than others, although there are also studies that include Daptomycin, Amikacin, Gentamicin, Ceftriaxone, Vancomycin, etc. It is noteworthy to mention that the authors of these reviews never indicate the type of gelatin (type A or B) used for sealing and how this affects the properties of the graft.

In this paper, we studied woven vascular grafts sealed with GelA or GelB containing one of the three antibiotics (Rifampicin, Ceftriaxone, or Vancomycin) in the sealing films. It has been shown that different combinations of these components affected the mechanical and antibacterial properties differently, as well as the cytocompatibility of grafts.

## 2. Results

We used gelatin type B from bovine skin, gelatin type A from porcine skin, Vancomycin, Ceftriaxone, and Rifampicin (see Section 4 for details).

### 2.1. General Characteristics 

#### 2.1.1. Water Permeability 

Our gelatin sealing technique ensured graft impermeability (Table 1) regardless of the type of gelatin used (A or B). The control group without sealing demonstrated a rather low permeability, only about 80 mL/min/cm^2^. This was largely due to the density of the plain weave (Figure 1A,B). Closing the pores with both gels resulted in complete impermeability to water (Table 1) due to the appearance of a gelatin film that sealed the pores between the fibers and fiber bundles in the woven fabric (Figure 1C,G).

When antibiotics were added to the gelatin sealing mixture, the gelatin film became visually less bulky (Figure 1D–F,H–J). However, a significant increase in permeability was observed only in the “gelatin B + Vancomycin” and “gelatin B + Rifampicin” grafts, and the permeability rates were eight (*p* = 0.003) and 12 (*p* = 0.0001) times lower than in the control group. “Gelatin A + Vancomycin” grafts showed a minimal increase in permeability compared to counterparts treated with gelatin A alone.

#### 2.1.2. Kinking Radius 

Both types of gelatins significantly increased the rigidity of the grafts, with type A gelatin being more pronounced. Gelatin A increased the kinking radius by 4 times (*p* = 0.00002), whereas gelatin B increased it only by 1.7 times (*p* = 0.001) (Table 2). Most likely, this is due to the difference in gel strength between gelatins A and B: ~300 g for type A gelatin and ~225 g for type B gelatin [27].

### 2.2. Antibacterial Properties

The disc diffusion method demonstrated high antimicrobial activities of both GelA and GelB with Vancomycin alone. These samples were active against St.aureus and Ent.faecalis, demonstrating wide zones of inhibition. The other samples did not inhibit the growth of the microorganisms at all (Table 1 and Figure 2).

### 2.3. Cytocompatibility 

All cytocompatibility tests were performed using EA.hy926 cells (see Section 4.6 for more details).

#### 2.3.1. Indirect Cytotoxicity 

The viability of EA.hy926 cells after 72 h of incubation in sample extracts varied from 99% to 121% (respectively, for samples No. 6 and No. 1). Cell viability of the control sample extracts (group No. 0) was 108% (Figure 3). Thus, according to [28], none of these extracts was cytotoxic for EA.hy926 cells within 72 h of observation. The significance of the differences between samples is presented in Appendix A.

#### 2.3.2. Cell Adhesion 

After 3 days of cultivation, the largest number of EA.hy926 cells was found in sample Nos. 2–4 (Figure 4). For sample No. 0 and 5–8, the number of cells was smaller but comparable to each other. After 7 days, an increase in the number of cells was detected in sample Nos. 2–8 (*p* < 0.05). A non-significant decrease in the number of attached cells was noted for sample No. 0 and No.1. On the 14th day, we observed a significant decrease in the number of adhered cells for all studied samples. For sample No. 0 and No. 1, the number of attached cells was the smallest. It should be noted that for sample No. 0, the number of attached EA.hy926 cells was the smallest over the entire observation period. The significance of the differences between samples in cell numbers after 3, 7, and 14 days of incubation is presented in Appendix A.

#### 2.3.3. Cell Viability 

After 3 days of cultivation, the viability of EA.hy926 cells in sample Nos. 1–4 and No. 6 was higher than that of control sample No. 0 (Figure 5). The viability of EA.hy926 cells in sample No. 0–4 significantly (*p* < 0.03) decreased on the 7th day of cultivation, whereas for sample No. 5–8, it did not change significantly. On the 7th day, cell viability in all samples sealed with the gelatins was higher compared to the control.

For most of the samples (Nos. 0–1 and 3–5), cell viability on the 14th day of cultivation significantly decreased compared to that on the 3rd (*p* < 0.04) day of cultivation, and it decreased for sample No. 1 and No. 5 compared to that on the 7th day (*p* < 0.03). However, after 14 days of cultivation, the viability of endothelial cells in the studied samples was higher compared to the control (sample No. 0). It should be noted that for sample Nos. 6–8, cell viability did not change significantly throughout the entire observation period. Cell viability in control sample No. 0 was the lowest throughout the entire observation period: after 3 days—73%, after 7 days—44%, and after 14 days—50% (Figure 5). The significance of the differences between samples in cell viability after 3, 7, and 14 days of incubation is presented in Appendix A.

#### 2.3.4. Cell Distribution 

To demonstrate the surface distribution and viability of the cells, they were stained with fluorescent dyes. Figure 6 shows that on day 3, the number of endothelial cells was significantly lower compared to day 7, but living cells (green) were predominant. The location of the cells throughout the observation period was oriented in the direction of the fibers. By the 7th day of cultivation on the samples, not only the total number of cells but also the number of dead cells (ethidium bromide, red color) increased significantly. It is likely that by day 7, gelatin desorption occurred, and the cells penetrated the samples. Figure 6 shows that on days 7 and 14, the cells were located at different depths, which impeded microscope focusing and cell counting. By the 14th day of cultivation, the number of endothelial cells on the surface of the samples decreased, and practically no dead cells were observed. It is likely that all dead cells came off the surface of the samples and were removed when the growth medium was replaced during cultivation. In some photographs, only a green glow of the fibers is visible, which may be due to the staining of the cells lying in the deeper layers of the samples. This green fluorescence of the fibers cannot be due to background noise or autofluorescence since samples not seeded with cells do not show fluorescence (Appendix A). Interestingly we found that endothelial cells demonstrate tube formation on samples coated with gelatin B + antibiotics (Nos. 6–8) (Appendix A). The spontaneous formation of tubules–capillary-like structures is a qualitative sign of the angiogenic activity of endothelial cells.

### 2.4. SEM and EDS Identification of Graft/Cell Interaction

According to fluorescence microscopy data, it seems that the cells began to penetrate the thickness of the prosthesis wall by the 14th day of cultivation. We tried to find evidence of this using a different independent method. For this purpose, an SEM with EMF was chosen.

Since the prosthesis itself was woven from polyester fiber, its main elements were C and O. Mapping of these elements revealed the pattern of the prosthesis (Figure 7, bottom row). Therefore, these elements cannot be used for imaging cells.

The gelatin film is well visualized using Na, Cl, and N. The latter is contained in large quantities in gelatin since gelatin is a derivative of the collagen protein (Figure 8). We performed an additional study with reference cells (L929 mouse fibroblast cell line) and found that the cells can be well visualized using Cl and Na in a media containing no Cl and Na (Appendix A). However, if the cell environment contains a lot of Cl and Na, as in our case, it is not possible to identify cells by these elements (Appendix A) since the result will be doubtful. 

We believe that it would be most appropriate to identify cells using typical intracellular elements: K, P, and S. This approach required a rather long scanning to obtain an adequate map since the amount of these elements is very small compared to C, O, Na, Cl, and N (Appendix A). In addition, the potassium and sulfur maps are very diffuse, but P can reliably reflect the localization of the cell (Appendix A). Using this approach, we visualized a large number of cells located along the polyester fibers in the deep layers of the graft wall.

## 3. Discussion

Gelatin is a derivative of collagen, the predominant protein of connective tissue. The first significant rise of interest in the properties of gelatin occurred in the 1920s [29,30] and then in the 1950s [29,31] of the last century. Currently, there is a new rise of interest in gelatin and its properties, associated with the rapid development of regenerative medicine and tissue engineering. 

However, when analyzing the literature, it may be surprising to find that basic research on gelatin properties is seldom taken into account by developers and manufacturers of medical devices, for example, vascular prostheses. The technology for sealing woven vascular grafts with gelatin has been known at least since the 1970s of the last century, but published works usually do not include information about the type of gelatin used for processing. This is true for both clinical [32] and experimental [23] studies. 

Meanwhile, it is well known that the properties of gelatin A and gelatin B are different, which is associated with their manufacturing technology. Gelatin A is obtained from porcine skin by acid hydrolysis, resulting in an isoelectric point (pIE) of about 9.0, and is called basic gelatin. Gelatin B (acidic, pIE = 5.0) is obtained from bovine skin or bones via alkaline hydrolysis [14,33,34]. Gelatins A and B differ radically in terms of hydrodynamic radius, viscosity, linear viscoelastic range, equilibrium degree of swelling, etc. [33,34]. Acid gelatin sorbs and desorbs bioactive molecules more easily and is, therefore, more often used to induce neoangiogenesis [35,36]. Gelatin B is also preferred for bioprinting [37]. 

In this work, we demonstrate for the first time that type A and type B gelatins have different effects on most properties of woven vascular grafts, despite the concentrations and sealing technologies being the same.

This first concerns the mechanical characteristics. Both A and B types of gelatin (without the addition of antibiotics) ensure the graft wall with complete water impermeability. However, when treated with gelatin A, the grafts become much more rigid; the kinking radius increases by four times. This parameter also increases for grafts sealed with GelB, but it is 2.5 times less than when treated with GelA.

When antibiotics are introduced into the sealing mixture, the graft properties also depend on the type of gel used. When using GelB, the permeability of the vessel increases more significantly than when using GelA, although it remains 7–13 times lower than that of the control. Gelatin films become thinner and “lighter” with the introduction of all antibiotics in both GelA and GelB sealants, but the combinations of gelatin B with Vancomycin and Ceftriaxone increases the graft kinking radius. At the same time, the kinking radii are reduced in the GelA counterparts. Therefore, adding these antibiotics to GelA helps to reduce its rigidity and GelB to increase it. Therefore, the dependence of the “Gel/antibiotic” interaction depends on the gel type. The addition of rifampicin decreases the kink resistance of all the grafts, regardless of the gelatin base type. Thus, the first conclusion from this study is that sealing with GelB reduces the rigidity of grafts with satisfactorily low permeability.

In our experiments, only Vancomycin provided antibacterial properties, regardless of which gelatin it was combined with. There are many studies that show the antibacterial effect of Rifampicin or Ceftriaxone [23,24], but we did not find such an effect.

It is known that the interaction between the vascular graft and endothelial cells is of the greatest importance. The long-term fate of any implant depends on the tissue reactions to them. The less cytotoxic the vascular implant, the faster the new *vasa vasorum* forms in its pores, and the less pronounced the neointimal hyperplasia [38]. In turn, the thicker the neointima layer, the higher the risk of thrombosis and stenosis of the implanted graft [39]. In this regard, we paid great attention to studying the possible relationship between the composition of the sealant and the cytocompatibility of the graft. 

In our work, we proceeded from the hypothesis that the cytotoxic effect may be associated primarily with the antibiotic. As the gel film swells and degrades, the antibiotic is released and cytotoxicity should decrease.

It turned out that this is not entirely true. First, when evaluating indirect cytotoxicity, the extracts of all samples did not have a toxic effect on EA.hy926 cells, although the extraction lasted for 3 days and should have been accompanied by the release of the antibiotic into the medium. Previous studies have proven that the biologically active molecules included in the gel films are completely released in a short time, from several hours to several days (up to 8 days), and the release from the GelB film is faster [33,34,35,36]. Thus, we conclude that none of the compositions chosen for the tested gel films exhibited indirect cytotoxicity.

Second, the worst adhesion and viability of EA.hy926 cells in direct contact with the samples were observed in the control group containing no gelatin film with or without antibiotics. Moreover, the longer the follow-up period, the lower the rates of both adhesion and viability in this group. The best adhesion on day 3 was observed for all samples with GelA films. On day 7, regardless of the composition of the film (samples 2–8), a sharp increase in the number of adherent cells was characteristic, with a decrease in the proportion of viable cells. By the 14th day, the number of adherent cells sharply decreased, whereas most of them remained viable (Figure 4, Figure 5 and Figure 6). We believe that by the 14th day, all dead cells “fall off” from the surface, and only living cells remain, which is clearly visible in Figure 6. In addition, a significant part of the cells, especially in groups 6–8 (gelatin B + antibiotics), migrated into the deep layers of the wall by day 14, which was almost impossible to detect using a fluorescent microscope, but was confirmed by SEM and EDS data (Figure 9). These cells were visualized (day 7) on the surface as some closed circular structures (Figure 6 and Appendix A), and they are dispersed in the deep layers of the wall along the polyester fibers of the graft. This allowed us to conclude that there are signs of spontaneous organization of endothelial cells into capillary-like structures even in the culture. Perhaps in vivo, this may accelerate the formation of *vasa vasorum*, which is a positive property.

Thus, based on our results, we concluded that the Vancomycin-added GelB film is the best choice for sealing woven polyester grafts.

## 4. Materials and Methods 

### 4.1. Materials 

Woven vascular prostheses were fabricated from 74dtex S110 polyethylene terephthalate fibers (Gruschwitz Textilwerke AG, Leutkirch, Germany) in the Science and Technology Park of BNTU “Polytechnic” (Minsk, Republic of Belarus). They all had a plain weave pattern, R_warp_/R_weft_ = 2/2. 

The following reagents were used in the study: Gelofusine 4% intravenous solution (B.Braun Medical, Melsungen AG, Switzerland), type B gelatin from bovine skin (NoG9391, Sigma Aldrich, St. Louis, MO, USA), type A gelatin from porcine skin (NoG2500, Sigma Aldrich, USA), isopropyl alcohol (Vecton, St. Peterburg, Russia), glycerol 99.0–101.0% (PanReac AppliChem, Barcelona, Spain), and glutaric aldehyde 25% (Panreac, Spain).

To increase bacterial resistance, the following antibiotics were used: Vancomycin (Pharmconcept, Redkino, Russia), Ceftriaxone (F.Hoffmann-La Roche Ltd., Basel, Switzerland), and Rifampicin (Ferain, Electrogorsk, Russia). 

### 4.2. Treatment Groups 

In the first stage, an 8% gelatin solution was prepared from distilled water and dry gelatin type A or type B. To achieve this, the mixture was thoroughly stirred at 24–26 °C for 10–15 min, and then kept for 24 h at 37 °C for the gel to swell. The next step was the addition of 4% succinilated gelatin Gelofusin to obtain a 6% gelatin solution. 

Tubular graft samples were washed with a 3% hydrogen peroxide solution, dried, and subjected to low-temperature sterilization with ethylene oxide (Steri-Vac 8XL, 3 M). Then, each fixed sample was loaded into a container filled with 750 mL of gelatin type A or B solution with or without one of the following antibiotics: (1)Vancomycin 3.3 g/L;(2)Ceftriaxone 3.3 g/L;(3)Rifampicin 1 g/L.

To form a gelatin film, the samples were exposed for 30 min in a sealing solution and then dried for 24 h at ambient temperature under aseptic conditions (5 ISO). 

The dried samples were immersed for 8 h in a solution of glutaraldehyde (0.02%), glycerol (15%), and isopropyl alcohol (75%) prepared with sterile distilled water for the chemical crosslinking of gelatin. After the crosslinking was completed, the samples were washed five times for 7 min in a 15% solution of glycerol in distilled water. The treated prostheses were dried under aseptic conditions at ambient temperature. 

All tests were performed on 9 control and sealed groups of prostheses. The tested properties and sample numbers are listed in Table 2. 

Crimped woven grafts were used for kinking radius and permeability measurements: the whole prostheses for the first test and the cut pieces for the second test. The 2–3 cm tubular pieces were cut off from the edge of the vessel and incised along the wall. Round samples with a diameter of 5 mm were cut from each non-corrugated prosthesis to study their antibacterial properties; for cell cultivation, the diameter was 9 mm. Precision cutting was performed using the MELAZ-Cardio laser device (Institute of Laser Physics SB RAS, Novosibirsk, Russia).

### 4.3. Integral Water Permeability 

Permeability was evaluated based on the flow rate of water through a given area of the prosthetic sample under a given hydrostatic pressure, according to [40]. A customized unit (Figure 10) of the hydrodynamic testing machine (MedEng, Penza, Russia) was used. This unit included a sample fixation device, a low-pressure transmitter with an accuracy of 0,5% of span (DMP 331 BD Sensors, Moscow, Russia), a digital pressure control unit, an oil-free air compressor, a 10 L water tank, and a fluid collection device. 

Each tested sample was hermetically fixed with rubber gaskets in the device after maximal stretching of the corrugated folds (Figure 10b); the cross-sectional area of the device aperture for fixing the samples was 0.78 cm^2^. The system hydraulic pressure was gradually raised to a constant level of 120 ± 2 mmHg and controlled on the monitor of the testing machine, and then the volume of liquid passing through the wall of the prosthesis in 1 min was recorded using a measuring container. Each sample was tested in 5 repetitions. The tests were carried out at room temperature. After the end of the tests, the rate of water leakage per 1 cm^2^ was calculated using the following formula: $$Water\;permability=\frac{Q}{A}$$
where *Q* is the flow rate of the sample (mm/min); 

*A* is the cross-section area of the aperture in the sample holder (cm^2^). 

### 4.4. Kinking Radius Measurements 

Kinking radius measurements were made using cylindrical calipers of known size with radii ranging from 4 to 42.5 mm in 1.5 mm increments, according to ISO 7198:2016 (Appendix A). A loop was formed from the test prosthesis, inside which a cylindrical caliper of maximum diameter was placed, and the ends of the prosthesis were tightened in opposite directions to reduce the loop (Appendix A), followed by replacement with a caliper of a smaller diameter. The tests were repeated until a kink or deformation appeared, reducing the lumen of the prosthesis. The value of the kink radius was considered the size of the caliper, using which led to the appearance of a kink or deformation of the prosthesis. 

### 4.5. Antibacterial Properties 

This study was carried out using the modified Kirby–Bauer disk diffusion test. We used Mueller–Hinton agar (Bio-Rad, Mitry-Mory, France) and two bacterial strains: *Enterococcus faecalis* (ATCC 29212) and *Staphylococcus aureus* (ATCC 29213). Bacterial inoculum was prepared by diluting the broth culture to match a 0.5 McFarland turbidity standard, which is equivalent to approximately 150 million cells per mL. Using an aseptic technique, each microorganism suspension was streaked across the agar surface, and then 4–6 samples of prosthetic disks from each group were applied to each Petri dish. All Petri dishes were incubated for 24 h at 36 °C, after which the zones (diameter, mm) of microorganism inhibition were measured. 

### 4.6. Cytocompatibility Evaluation 

#### 4.6.1. Cell Culture

The EA.hy926 endothelial cells were kindly provided by Dr. C. J. Edgell (Carolina University, Winston-Salem, NC, USA). The EA.hy926 cells were cultured in DMEM/F12 (Gibco, Carlsbad, CA, USA) medium supplemented with 10% fetal calf serum (FCS; Hyclone, Logan, UT, USA), 2 mM L-glutamine (ICN, Costa Mesa, CA, USA), and 40 μg/mL gentamicin sulfate (Dalkhimpharm, Khabarovsk, Russia) in a humid atmosphere with 5 vol% CO_2_ at 37 °C until a confluent monolayer was formed. Cells were cultured in flasks with the culture medium replaced every 3 or 4 days and removed using Trypsin-Versene (Biolot, Saint Petersburg, Russia) during passaging.

#### 4.6.2. Indirect Cytotoxicity of Samples

For preparing the extracts, flat samples (9 mm in diameter) were incubated in complete growth medium (DMEM/F12 supplemented with 10% FCS, 2 mM glutamine, and 40 μg/mL gentamicin) for 72 h at 37 °C and a surface area/volume ratio of 1.25 cm^−1^. Subsequently, the samples were removed from the medium, and the extracts were used to determine cytotoxicity against EA.hy926 cells.

EA.hy926 cells were seeded in a 96-well cell culture plate at 1 × 10^4^ cells per well and incubated for 24 h to allow attachment. Then, the medium was removed, and 100 μL of the extract was added, followed by cultivation for another 72 h. Cell viability was then determined using the 3-(4,5-dimethylthiazol-2-yl)- 2,5-diphenyl-2H-tetrazolium bromide (MTT) assay (Sigma-Aldrich, Darmstadt, Germany) according to the manufacturer’s instructions. MTT (5 mg/mL) was added to each well, and incubation was continued for another 4 h. Formazan crystals were dissolved in 150 μL dimethyl sulfoxide (PanReac AppliChem, Darmstadt, Germany). The absorbance of the dissolved formazan crystals was measured at 492 nm using a Stat Fax-2100 multiwall plate reader (Awareness Technology Inc., Palm City, FL, USA). Cell viability was calculated using the following equation: (A experimental group/A control group) × 100%, where A is the optical density of the samples. Intact cells cultured in DMEM/F12 medium without the samples were used as the control group.

#### 4.6.3. Cell Adhesion and Viability 

EA.hy926 cells were seeded at a density of 5 × 10^4^ cells on the samples and cultured for 3, 7, and 14 days in a 24-well cell culture plate. The culture medium was replaced every 3 or 4 days. At each time point, the samples were rinsed twice with sterile phosphate-buffered saline (PBS) and transferred to new 24-well tissue culture plates. Then, the cells were removed from the samples using Trypsin-Versene. After contact with the samples, the cell count and their viability were measured using a Countess II Automated Cell Counter (Invitrogen, Carlsbad, CA, USA) after staining with 0.4% trypan blue (Invitrogen, USA).

#### 4.6.4. Cell Distribution

The samples were placed in the wells of a 24-well plate and exposed to cells (5 × 10^4^ cells per sample) followed by cultivation for 3, 7, and 14 days. The cell distribution was determined by staining with fluorescent dyes: acridine orange (DIA M, Russia; 100 μg mL^−1^) and propidium iodide (Medigen, Novosibirsk, Russia; 100 μg mL^−1^). The samples were then incubated for 10 min at 37 °C and examined using an Axio Observer microscope (Zeiss, Oberkochen, Germany). 

### 4.7. Scanning Electron Microscopy (SEM) and Energy Dispersive Spectrometry (EDS) Analysis 

Each sample was straightened, fixed, and dried at room temperature under sterile conditions. Before testing, the samples were coated with a 25–30 nm thick conductive carbon layer using a GVC-3000 Thermal Evaporation Carbon Plating Instrument (KYKY Technology Co., Ltd., Beijing, China). 

SEM and EDS analyses of the cell-containing samples (7 and 14 day culture) and elemental mapping of selected areas were carried out using a WIN SEM A6000LV scanning electron microscope (KYKY Technology Co., Ltd., Beijing, China) equipped with an EDX system AzTec One (Oxford Instruments, Bristol, UK). Sample observation was performed using secondary electron (SE) and back-scattering electron (BSE) detectors at a high electron voltage of 20 keV and an electron beam setting of 120 µA. Ten fields of observation were selected and examined at 100×, 250×, 450×, and 700× magnification for each sample. The sizes of the observed objects were measured using the KYKY SEM Microsoft operating program.

### 4.8. Statistical Analysis

Quantitative data were processed using Dell Statistica 13.0 (Dell Software Inc., Round Rock, TX, USA) and presented as mean (M) and standard deviation (±SD). The normality of the data distribution was determined using the Shapiro–Wilk test. The nonparametric Mann–Whitney U-test was used to compare the two groups, and the level of significance was set to *p* < 0.05. 

## 5. Conclusions

On evaluating the results obtained as a whole, we concluded that the optimal choice is the sealing technology with GelB and Vancomycin. This combination provides satisfactory mechanical characteristics (permeability~10 mL/min/cm^2^, kinking radius 21 mm) and good antibacterial properties (inhibition zones for *Staphilococcus aureus*~15 mm, for *Enterococcus faecalis*~12 mm). The samples of grafts sealed with “GelB + Vancomycin” did not exhibit indirect or direct cytotoxicity. By day 7, the cells in these samples began to self-organize into capillary-like structures. 

## 6. Study Limitations 

1. In this work, only one type of prosthesis was studied. We have previously shown that the mechanical properties of woven grafts largely depend on the weaving technologies [11]. Therefore, we cannot exclude the possibility that changes in the graft structure itself (density of fibers and fiber bundles, type of weaving or knitting, etc.) may lead to changes in the interaction of the graft with gelatins.

2. We studied only three drugs belonging to different classes of antibiotics. A separate study is necessary to compare the effectiveness of various antibiotics added to the gelatin film, the ratio of their bacteriotoxicity and cytotoxicity, incorporation into the gelatin film, and release from it. It may be advisable to apply computational pharmaceutical approaches to model the interactions of different antibiotics with different gelatin types. 

3. The tendency of EA.hy926 endothelial cells to self-organize capillary-like structures when interacting with the films of the “GelB + Vancomycin” composition that we discovered needs further study using other in vitro and in vivo models and other types of endothelial cells (HUVEC, ECP, HAEC, etc.)

## Figures and Tables

**Figure 1 ijms-25-00965-f001:**
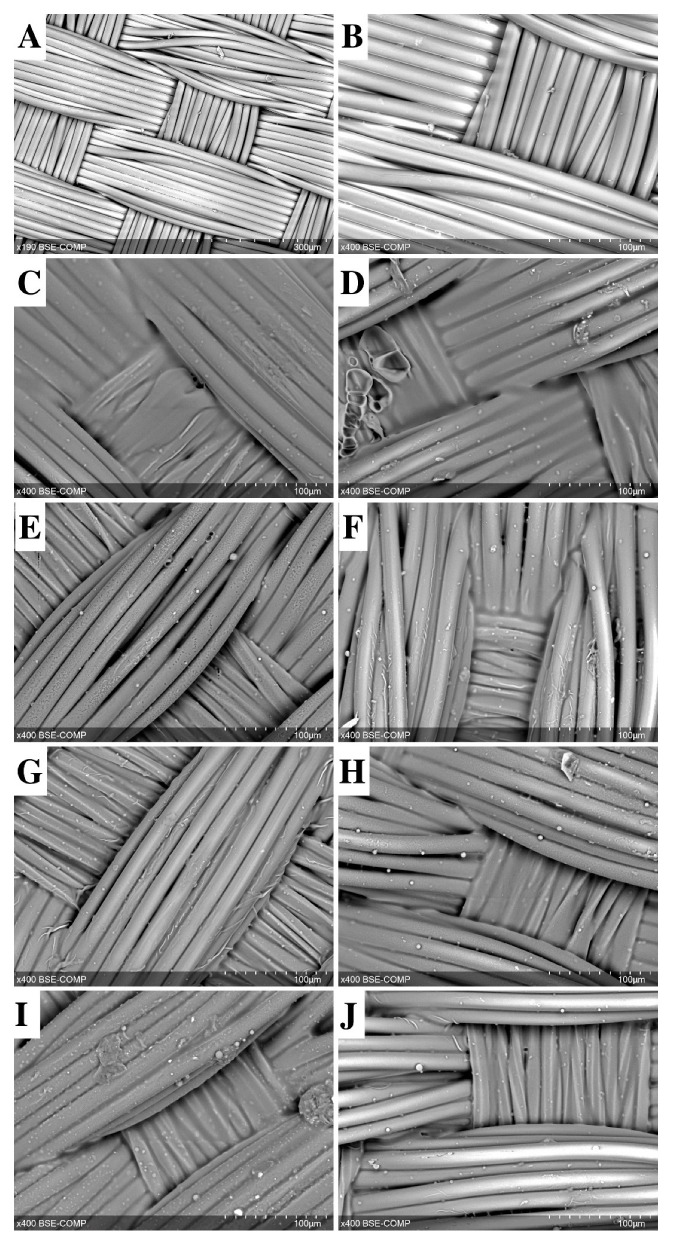
SEM image of a non-sealed vascular woven graft (**A**,**B**) sealed with gelatin A (**C**,**E**,**G**,**I**) or gelatin B (**D**,**F**,**H**,**J**). Gelatin film with antibiotics: Vancomycin (**E**,**F**), Ceftriaxone (**G**,**H**), and Rifampicin (**I**,**J**). Scale bars are 300 μm (**A**) and 100 μm (**B**–**J**).

**Figure 2 ijms-25-00965-f002:**
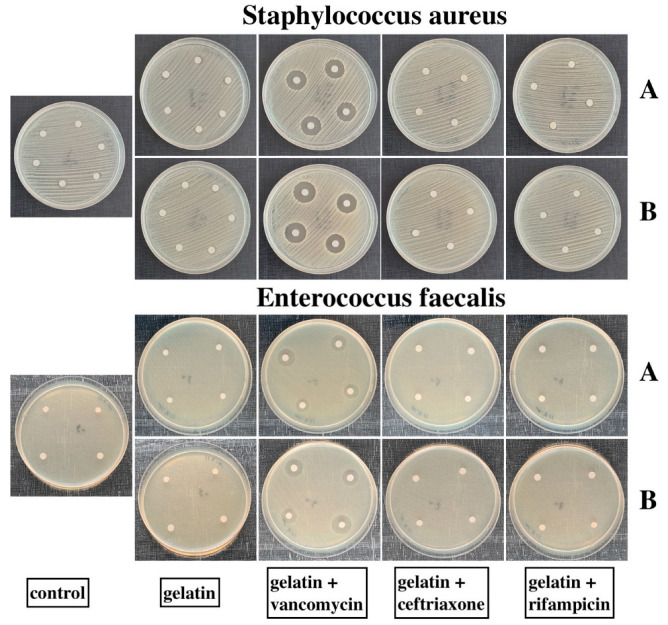
Petri dishes containing St.aureus and Ent.faecalis cultures. “A” rows are the samples sealed with GelA, “B” rows are the samples sealed with GelB. No inhibition zone was observed on the dishes with the control, gelatins, gelatins + Ceftriaxone, and gelatins + Rifampicin samples. GelA and GelB + Vancomycin samples suppressed the growth of both St.aureus and Ent.faecalis.

**Figure 3 ijms-25-00965-f003:**
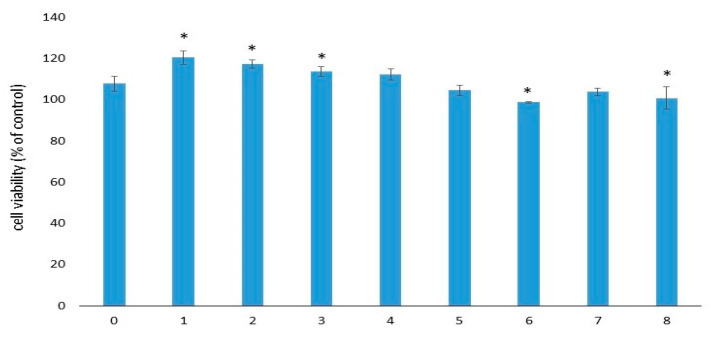
Viability of EA.hy926 cells after 72 h of incubation with the extracts. Data (*n* = 6 in each group) are presented as mean ± SD. The control group consisted of cells cultivated in complete DMEM/F12 medium without samples. Group No. 0 included the control samples without gelatins and antibiotics. *—*p* ≤ 0.05 compared to sample No. 0.

**Figure 4 ijms-25-00965-f004:**
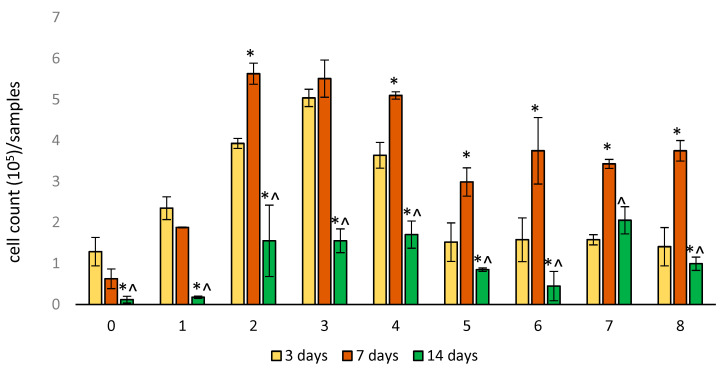
Cell adhesion on samples of woven vascular prostheses sealed in different ways. The numbering of groups corresponds to Table 1. Data (*n* = 5 in each group) are presented as mean ± SD. *—*p* < 0.05 compared to 3 days; ^—*p* < 0.05 compared to 7 days.

**Figure 5 ijms-25-00965-f005:**
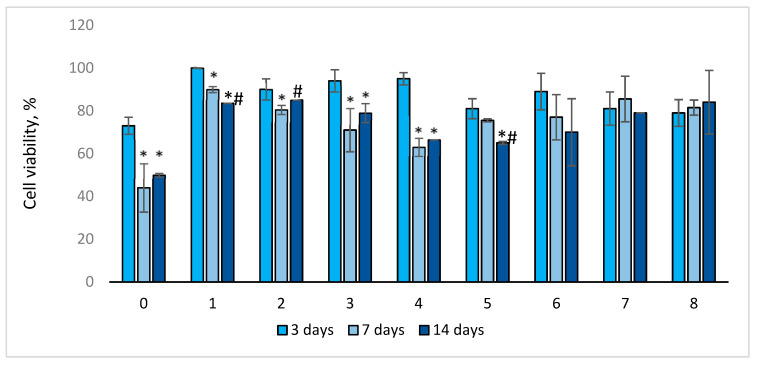
Viability of EA.hy926 cells depending on the days of sample cultivation. Trypan blue staining is presented as mean ± SD a (*n* = 5 in each group). *—*p* < 0.05 compared to 3 days; #—*p* < 0.05 compared to 7 days.

**Figure 6 ijms-25-00965-f006:**
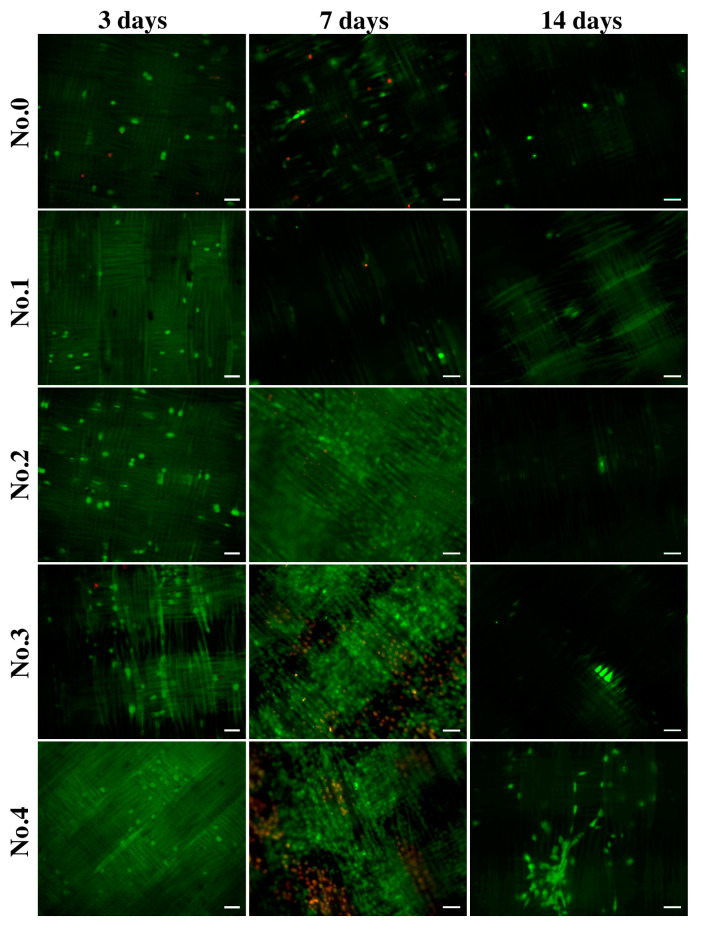
Fluorescence microscopy images of EA.hy926 cells on gelatin-coated samples after three, seven, and fourteen days of culture. Staining was performed using acridine orange (green, live cells) and propidium iodide (red, dead cells). The scale bar is 50 µm.

**Figure 7 ijms-25-00965-f007:**
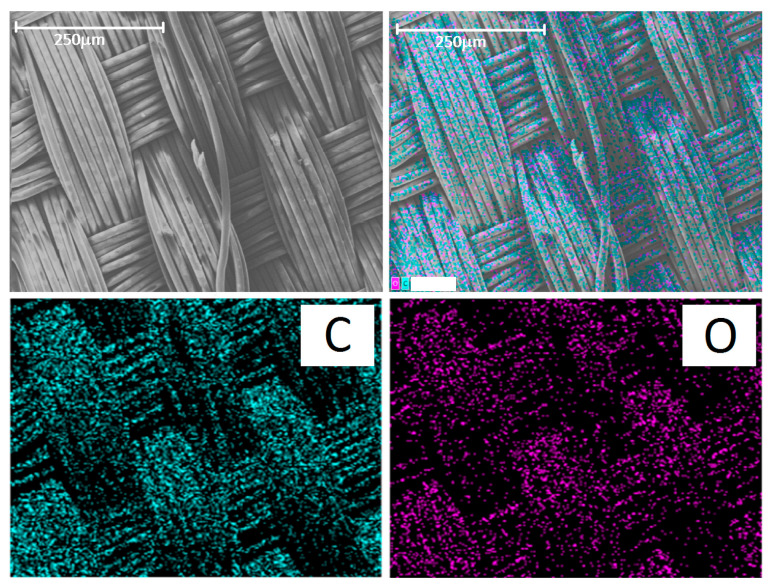
(**Top row**): SEM image of the control group prosthesis and the merged image. (**Bottom row**): distribution of carbon (C) and oxygen (O). The scale bar is 250 μm.

**Figure 8 ijms-25-00965-f008:**
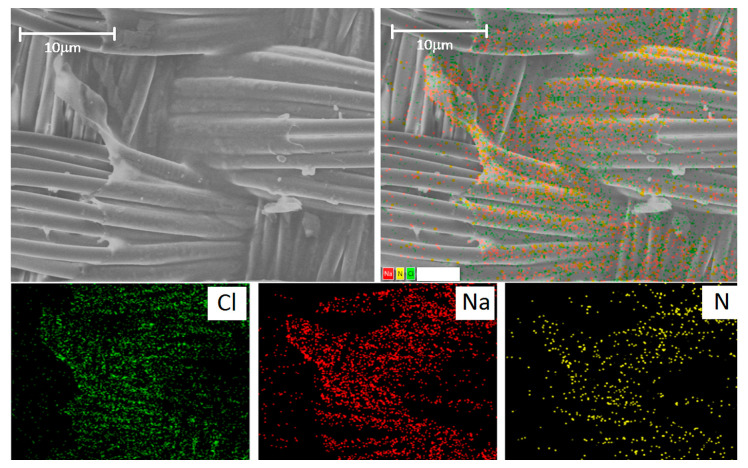
(**Top row**): SEM image of the gelatin-sealed graft and the merged image. (**Bottom row**): distribution of chlorine (Cl), sodium (Na), and nitrogen (N). The scale bar is 100 μm.

**Figure 9 ijms-25-00965-f009:**
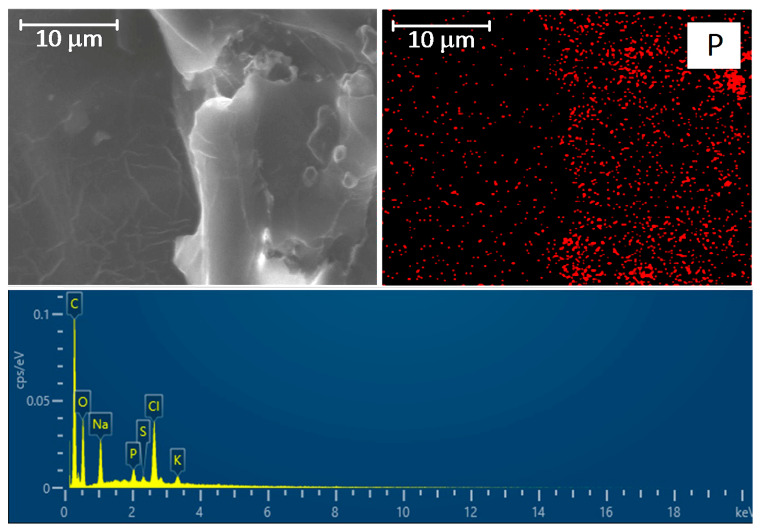
(**Top row**): SEM image of the graft wall cross-section and distribution of phosphorus (P). The scale bar is 10 μm. (**Bottom row**): EDS spectrum of the observation field.

**Figure 10 ijms-25-00965-f010:**
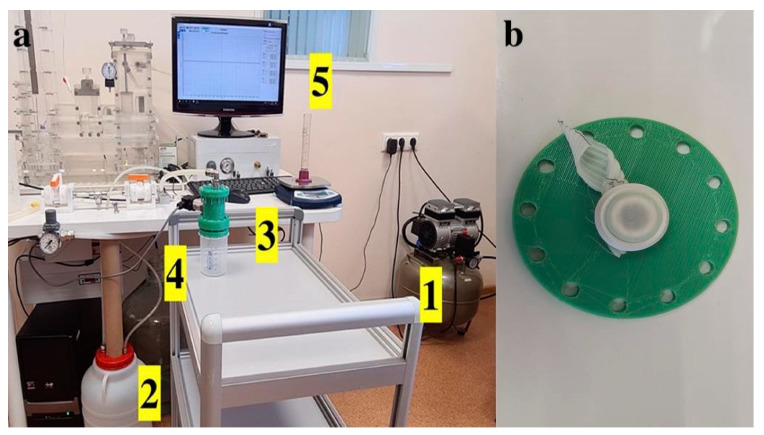
(**a**) Permeability testing according to ISO 7198:2016(E). 1—air compressor, 2—tank for distilled water, 3—sample fixation device, 4—water pressure sensor installed in the fixation device, and 5—pressure information is displayed on the monitor. 1 (**b**) Fixation of the test sample in a straightened state with a rubber rim into the 0.78 cm^2^ aperture.

**Table 1 ijms-25-00965-t001:** Main mechanical and microbiological characteristics of the tested woven grafts.

Group No.	Sample Treatment	Characteristics			
		Kinking Radius, mm	Water Permeability, mL/min/cm^2^	m/o Inhibition Zone, mm *	
				St. Aureus	Ent. Faecalis
0	Control (−)	10.9 ± 2.8	78.8 ± 2.7	0	0
1	GelA	42.5 ± 4.0	0	0	0
2	GelA + Vancomycin	21.0 ± 0	2.1 ± 0.7	15.02 ± 0.73	12.52 ± 2.12
3	GelA + Ceftriaxone	25.0 ± 0	0	0	0
4	GelA + Rifampicin	51.0 ± 0	0	0	0
5	GelB	17.3 ± 0.4	0	0	0
6	GelB + Vancomycin	21.0 ± 0	10.8 ± 6.1	15.81 ± 0.45	12.38 ± 1.49
7	GelB + Ceftriaxone	34.9 ± 0.75	0.2 ± 0	0	0
8	GelB + Rifampicin	19.0 ± 0	6.1 ± 0.4	0	0

* m/o—microorganism; disk diffusion method was used.

**Table 2 ijms-25-00965-t002:** Sample numbers tested in different experiments.

Group No.	Sample Treatment	Kinking Radius	Water Permeability	Antibacterial Properties	Cytocom- Patibility	SEM and EDS
0	Control (−)	12	6	20	24	8
1	Gelatin A	4	6	20	24	8
2	Gelatin A + Vancomycin	4	6	20	24	8
3	Gelatin A + Ceftriaxone	4	6	20	24	8
4	Gelatin A + Rifampicin	4	6	20	24	8
5	Gelatin B	4	6	20	24	8
6	Gelatin B + Vancomycin	4	6	20	24	8
7	Gelatin B + Ceftriaxone	4	6	20	24	8
8	Gelatin B + Rifampicin	4	6	20	24	8

## Data Availability

Data is available within the article.

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
