# Peer review of "Which Gelatin and Antibiotic Should Be Chosen to Seal a Woven Vascular Graft?"

_ijms, 2024, doi:10.3390/ijms25020965_

Round 1
Reviewer 1 Report
Comments and Suggestions for Authors
Here, the authors demonstrate the coating of large-diameter vascular grafts with different antibiotics. Extensive characterization of the coated graft has been performed to study water permeability, kinking, the effect of antibiotics, etc. Although the study is significant in clinical application, the manuscript has several flaws. See below some of the comments.
1) Lines 14 and 22: Define what authors mean by ‘Optimal’
2) The abstract needs to be rewritten. The language is casual, and the idea is not conveyed.
3) In the introduction, the importance of sustained release of antibiotics in impregnated is highlighted, but the authors failed to perform the release kinetics of vancomycin in impregnated graft. Note that the disc diffusion study will not be a substitute for the release kinetics study.
4) Both Gel A + vancomycin and Gel B + vancomycin hold similar properties except for water permeability. Why was Gel A + vancomycin not selected despite lower water permeability than Gel B + vancomycin?
5) Was the kink radius performed in the wet state? The coating material should have a negligible impact if the graft is moistened.
6) Why are the kink radius and water permeability values different in Table 1 and Table 2?
Comments on the Quality of English LanguageThe manuscript has a several of flaws and is confusing to the reader.
Author Response
Dear reviewer,
We thank you for your valuable comments and recommendations.
1. Lines 14 and 22: Define what authors mean by ‘Optimal’
The abstract is rewritten, “Optimal” was deleted
2. The abstract needs to be rewritten. The language is casual, and the idea is not conveyed.
We have edited the article and tried our best to improve the English language. The abstract is rewritten. Аll corrections are marked in yellow.
3) In the introduction, the importance of sustained release of antibiotics in impregnated is highlighted, but the authors failed to perform the release kinetics of vancomycin in impregnated graft. Note that the disc diffusion study will not be a substitute for the release kinetics study.
We agree with you, the disc diffusion study will not be a substitute for the release kinetics study. Moreover, it is necessary to evaluate not only the antibiotic release kinetics, but also the desorption kinetics of the gelatin sealant itself. However, in this article we aimed to evaluate the EFFECTS of various compositions of gelatins and antibiotics, but not the reasons for these effects. In the future, we plan to perform these kinetic studies with 2 compositions: GelA + vancomycin and GelB + vancomycin
4) Both Gel A + vancomycin and Gel B + vancomycin hold similar properties except for water permeability. Why was Gel A + vancomycin not selected despite lower water permeability than Gel B + vancomycin?
Our preference for «Gel B + vancomycin» sealant was based on several reasons. The first is the tendency of endothelial cells to form capillary-like structures in these samples. This is very important for grafts implanted in patients, since micro-vascularization of the graft wall prevents neointimal hyperplasia (Sánchez P.F., Brey E.M., Briceño J.C. Endothelialization mechanisms in vascular grafts. J Tissue Eng Regen Med 2018; 12(11): 2164–2178, https: //doi.org/10.1002/term.2747). Secondly, judging by the literature, GelB itself has greater cytocompatibility with different cell types. In addition, the mechanical properties of both sealed and non-sealed prostheses are highly dependent on weaving techniques and can be controlled through weaving rapports (Shadanov А.А., Timchenko Т.P., Vladimirov S.V., Lushchyk P.E., Zablotsky А.V., Kiselyov S.О., Zhuravleva I.Yu., Sirota D.А., Chernyavskiy А.M. The Influence of Weaving Technologies on the Integral Characteristics of Synthetic Vascular Prostheses. Sovremennye tehnologii v medicine 2022; 14(6) : 5-13, https://doi.org/10.17691/stm2022.14.6.01). Possibly, we can get good mechanical results with GelA. Of course, the interaction of both “GelA + vancomycin” and “GelB + vancomycin” coatings with endothelial cells will be studied further to obtain a graft with minimal permeability and kinking radius and maximal antibacterial activity and cytocompatibility.
5) Was the kink radius performed in the wet state? The coating material should have a negligible impact if the graft is moistened
You are absolutely right: the coating material should have a negligible impact if the graft is moistened. However, we carried out these tests according to the methods recommended by the ISO 7198-2016 standard. This standard does not provide for hydration of grafts before testing, since this introduces additional unknowns, for example, different hydration degrees of the test samples.
6) Why are the kink radius and water permeability values different in Table 1 and Table 2?
Table 1 shows the numerical values of the characteristics, while Table 2 shows the number of samples for each type of test.
Reviewer 2 Report
Comments and Suggestions for Authors
The manuscript by Zhuravleva et al. describes the effect on Gelatin A or B and 3 often used antibiotics on cell adhesion, viability and cytotoxicity of endothelial cells (EA.hy926), and antibacterial properties against Staphilococcus aureus and Entrococcus faecalis on woven grafts from PET for aortic protheses in vitro.
The research design is thorough, uses standard methods, and the story is written straight forward. The authors follow a clear thread, concentrating on the comparison of GelA and GelB and the cytotoxic effect of the three antibiotics Rifampicin, Ceftriaxone, and Vancomycin. The results of the different methods used suppport the drawn conclusion, i.e. that GelB + Vancomycin would be the optimal candidate of the tested variants for the aortic graft. The reviewer only points out some minor points to be optimized/corrected:
1. The fluorescent microscopy images of Fig. 6 are done in low resultion/magnification, and thus not clear in identifying the viability and abundance of the endothelial cells. Due to background noise of the samples, in particular the 14 d samples are hard to analyse. Moreover, the differences between the samples are not that prominent, despite the occurance of dead cells of day 7 samples.
2. EDS mapping of graft wall sections in Fig. 9 are not resolved enough, and thus rather show arbitrary distribution of P, K, and S atoms. The conclusion drawn from Fig. 9 are not well supported. Morevover, the counts of the EDS spectrum are comparatively low. Since the settings for the mapping are not defined in the Mat/Meth section 4.7 on p. 18, the reviewer suspect that either the signal was not optimal, or the collecting time was to short. Please comment on this, provide the settings for EDS mapping, and/or provide better resolved analysis.
3. SEM images in Fig. 1 display BSE detector, but in Mat/Meth section 4.7 on p. 18, the authors claim SE detector to be used. Please correct accordingly.
4. In the Ref. section p. 17ff, upper and lower case styles titles are mixed. Please correct accordingly.
Comments on the Quality of English Language
The quality of English language is high, but some errors in grammar and expression, as well as spelling do occur. Minor editing of the manuscript is necessary.
Author Response
Dear reviewer,
We thank you for your valuable comments and recommendations.
The reviewer only points out some minor points to be optimized/corrected:
1. The fluorescent microscopy images of Fig. 6 are done in low resultion/magnification, and thus not clear in identifying the viability and abundance of the endothelial cells. Due to background noise of the samples, in particular the 14 d samples are hard to analyse. Moreover, the differences between the samples are not that prominent, despite the occurance of dead cells of day 7 samples.
The fibers of the vascular woven graft lie at different depth, therefore, during microscopy at high magnification, focusing the microscope is difficult. This can be clearly seen in Figure S2 for images with a scale bar of 20 µm. In this regard, we did not use high magnification. We attempted to reduce background noise in the images of day 14 samples. Fluorescence microscopy images (Figure 6) is presented only to demonstrate the distribution of cells on vascular woven grafts. Moreover, figure 6 shows that on days 7 and 14 the cells are located at different depths, which impede microscope focusing and cell counting. Therefore, the cell count and their viability after contact with the samples were measured on a Countess II Automated Cell Counter after staining with 0.4% trypan blue (Figure 5). The decrease in the number of cells on the surface of the samples by day 14 may be due to the detachment of dead cells from the surface of the samples and their removal when the growth medium was replaced during cultivation, as well as the penetration of living cells into the depths of the samples.
2. EDS mapping of graft wall sections in Fig. 9 are not resolved enough, and thus rather show arbitrary distribution of P, K, and S atoms. The conclusion drawn from Fig. 9 are not well supported. Morevover, the counts of the EDS spectrum are comparatively low. Since the settings for the mapping are not defined in the Mat/Meth section 4.7 on p. 18, the reviewer suspect that either the signal was not optimal, or the collecting time was to short. Please comment on this, provide the settings for EDS mapping, and/or provide better resolved analysis.
Thank you for the good adviсe. We followed your recommendation and increased the collecting time to 2.5 hours and the voltage to 25 keV. To test this approach, we used a mouse fibroblast culture. It turned out that it is P that most reliably reflects the localization of the cell (Fig. S5), but the distribution of S and K remains more diffuse. Therefore, we changed Fig. 9 and slightly edited the text of section 2.4
3. SEM images in Fig. 1 display BSE detector, but in Mat/Meth section 4.7 on p. 18, the authors claim SE detector to be used. Please correct accordingly.
All the surface images were obtained using a BSE detector, and EDS analysis was performed using a SE detector. The Mat/Meth 4.7 section has been corrected.
4. In the Ref. section p. 17ff, upper and lower case styles titles are mixed. Please correct accordingly.
Thank you for bringing this error to our attention. We corrected it.
Comments on the Quality of English Language
The quality of English language is high, but some errors in grammar and expression, as well as spelling do occur. Minor editing of the manuscript is necessary.
We have edited the article and tried our best to improve the English language.
Reviewer 3 Report
Comments and Suggestions for Authors
The manuscript reports an analysis of the influence of different gelatin and antibiotics on the sealing properties of a vascular graft.
The reviewer thinks this work might be published only after a careful revision. Otherwise, the advancement within the state of the art remains unclear.
Major revisions:
- Abstract to be rewritten. Unclear about the hypothesis of the work and the scientific question to be answered
- Please revise the Introduction carefully. For example, line 60 reports a reference to antibacterial drugs, and then the description is referred to gelatin.
The reviewer also considers the addition of the following references to highlight the novel trends in vascular graft:
Natta, Lara, et al. "Soft and flexible piezoelectric smart patch for vascular graft monitoring based on aluminum nitride thin film." Scientific Reports 9.1 (2019): 8392.
Cafarelli, Andrea, et al. "Small-caliber vascular grafts based on a piezoelectric nanocomposite elastomer: Mechanical properties and biocompatibility." Journal of the Mechanical Behavior of Biomedical Materials 97 (2019): 138-148.
- Table 1 needs to be clarified. There are values without a standard deviation, and others are reported as 0. Please explain if such data were unavailable, or 0, and the number of samples analyzed; otherwise, they are not comparable.
- Please discuss why gelatin can increase the rigidity of the prosthesis.
- Clarify the statistical analysis performed in Figures 3 and 4. Why the p-values of reference were 0.02 or 0.027 in both cases instead of the most classical 0.05 and 0.01? Check also the statistics in Figure 5.
- Which is the reason for the lower cell count on day 14 in Figure 4? The authors justify the lower presence of cells with"It is likely that by day 7, gelatin desorption occurs and cells penetrate into the samples.". Please provide a more convincing justification for such a result.
- Please provide more explicit images of Live/dead in Figure 6. A few cells are visible in many cases, or the background signal has a high presence.
- The EDS analysis reports the presence of C and O, and the analysis was also performed on Cl, Na and N, then also on P, K, and S. Which component contains Cl? Are these elements all present in gelatin and in the antibiotics? The significance of this analysis needs to be clarified.
- This work needs to clarify the role of gelatin and the antibiotics over time. Do they dissolve or be released? Which is the kinetics? Please investigate this point to better highlight the manuscript findings.
- Revise the statistical analysis, reporting if data are normal or not, and the number of samples analyzed.
Minor revisions:
- Please explain the meaning of hybrid prostheses
- Please clarify the preparation procedure of the samples. Is the gelatin added to the Gelosufin? What is the final concentration of gelatin?
- In Table 2, please add the unit of measure. What are the differences with Table 1?
Comments on the Quality of English Languagenothing to add, just a general revision
Author Response
Dear reviewer,
We thank you for your valuable comments and recommendations.
Major revisions:
- Abstract to be rewritten. Unclear about the hypothesis of the work and the scientific question to be answered
Abstract is rewritten
- Please revise the Introduction carefully. For example, line 60 reports a reference to antibacterial drugs, and then the description is referred to gelatin.
We have edited this paragraph (Line 67-76) for clarity.
- The reviewer also considers the addition of the following references to highlight the novel trends in vascular graft:
Natta, Lara, et al. "Soft and flexible piezoelectric smart patch for vascular graft monitoring based on aluminum nitride thin film." Scientific Reports 9.1 (2019): 8392.
Cafarelli, Andrea, et al. "Small-caliber vascular grafts based on a piezoelectric nanocomposite elastomer: Mechanical properties and biocompatibility." Journal of the Mechanical Behavior of Biomedical Materials 97 (2019): 138-148.
As for citing articles recommended by the reviewer, they are, unfortunately, very far from the topic of our study.
- Table 1 needs to be clarified. There are values without a standard deviation, and others are reported as 0. Please explain if such data were unavailable, or 0, and the number of samples analyzed; otherwise, they are not comparable.
The number of samples used for each test is indicated in Table 2, section Materials and Methods. 0 in Table 1 has different meanings. If 0 is indicated as a standard deviation, then this means that all tested samples showed exactly the same values (this happens sometimes). If 0 is indicated for permeability values, it means that the grafts were non-permeable for water at all. If 0 is indicated for microorganism inhibition zones, this means that there was no such zone, that is, these samples do not have antibacterial properties. This is clarified in sections 2.1 and 2.2 of the Results.
- Please discuss why gelatin can increase the rigidity of the prosthesis.
The rigidity of textile prostheses is increased to a greater extent not by gelatin itself, but by those structures that are formed during its cross-linking, for which various cross-linking agents are used (M Madaghiele , A Piccinno, M Saponaro, A Maffezzoli, A Sannino. Collagen- and gelatine-based films sealing vascular prostheses: evaluation of the degree of crosslinking for optimal blood impermeability. J Mater Sci Mater Med, 2009;20(10):1979-89. doi: 10.1007/s10856-009-3778-1). It is impossible not to cross-link gelatin, since this gelatin gel swells and desorbs very quickly. To reduce the negative impact of cross-linkers on the gelatin film, plasticizers are used, glycerol in particular. However, this does not completely avoid an increase in rigidity, especially when the gelatin film dries. When hydrated, it is less rigid.
- Clarify the statistical analysis performed in Figures 3 and 4. Why the p-values of reference were 0.02 or 0.027 in both cases instead of the most classical 0.05 and 0.01? Check also the statistics in Figure 5.
Quantitative cell data were processed using Dell Statistica 13.0 (Dell Software Inc., USA) and presented as mean (M) and standard deviation (+SD). The nonparametric Mann-Whitney U- test was used to compare two groups. The significance level was set to p < 0.05. The reference p values in Figures 3, 4, and 5 are adjusted to the classic p values < 0.05.
- Which is the reason for the lower cell count on day 14 in Figure 4? The authors justify the lower presence of cells with"It is likely that by day 7, gelatin desorption occurs and cells penetrate into the samples.". Please provide a more convincing justification for such a result.
The decrease in the number of cells on the samples by day 14 may be due to several reasons. On the 7th day of cultivation, an increase in the number of dead cells was noted, which was not associated with the cytotoxicity of the samples. We did not detect cytotoxicity of sample extracts in the MTT test at extraction times of 3 days (Fig. 3) and 7 days (data not shown). Therefore, cell death is likely related to the surface morphology of the vascular woven graft, on which cells cannot fully spread and divide. By the 14th day of cultivation, dead cells detach from the surface of the samples and are removed when the growth medium is replaced during the cultivation, and the remaining living cells proliferate poorly. This leads to a decrease in the number of cells on the 14th day of cultivation.
- Please provide more explicit images of Live/dead in Figure 6. A few cells are visible in many cases, or the background signal has a high presence.
We tried to improve image clarity and reduce background noise in the images shown in Figure 6.
- The EDS analysis reports the presence of C and O, and the analysis was also performed on Cl, Na and N, then also on P, K, and S. Which component contains Cl? Are these elements all present in gelatin and in the antibiotics? The significance of this analysis needs to be clarified.
C and O are contained in large quantities in the polyester fiber itself; Cl, Na, and N are clearly visible in the gelatin film, so all these 5 elements cannot be used for cell identification. This is explained in section 2.4 of the Results. Cl and Na appear in the film due to Gelofusin (they are part of it). Cells in media that do not contain Cl and Na are also well visualized by Cl and Na (see Fig. S4). However, if the cell’s environment contains a lot of Cl and Na, as in our case, then it is optimal to visualize cells using typically intracellular elements: P, K and S (Fig.S5). In this case, P, as a rule, provides the best visualization (see Fig. S6), so we edited the Fig.9 and the text in section 2.4.
- This work needs to clarify the role of gelatin and the antibiotics over time. Do they dissolve or be released? Which is the kinetics? Please investigate this point to better highlight the manuscript findings.
Antibiotics are released from the gelatin gel within a fairly short time. The gelatin gel itself is gradually desorbed from the woven fabric, replaced by proteins, blood components, and the patient's cells. All these are known facts, and we did not focus on them, we only emphasized the difference between gelatins A and B (line 327-332). In this article we aimed to evaluate the EFFECTS of various compositions of gelatins and antibiotics, but not the reasons for these effects. In the future, we plan to perform these kinetic studies with 2 compositions: GelA + vancomycin and GelB + vancomycin.
- Revise the statistical analysis, reporting if data are normal or not, and the number of samples analyzed.
The number of samples analyzed is indicated in Table 2. We have edited its title to make its meaning clear without studying the contents of the table itself. We detailed the statistical analysis methodology (section 4.8). Quantitative cell data (n = 5 in each group) were analyzed using Dell Statistica 13.0 (Dell Software Inc., USA) and presented as mean (M) and standard deviation (+SD). The normality of data distribution was determined with Shapiro–Wilk test. The nonparametric Mann-Whitney U- test was used to compare two groups.The significance level was set to p < 0.05.
Minor revisions:
- Please explain the meaning of hybrid prostheses
We believed that to understand what a hybrid prosthesis is, a description of the FET technique for which it is used is sufficient (line 43-46). However, we agree that this may not be enough for non-specialists in this area, and we introduced the definition of a hybrid prosthesis in the new edition of the article (line 47-50).
- Please clarify the preparation procedure of the samples. Is the gelatin added to the Gelosufin? What is the final concentration of gelatin?
We added Gelofusin (4% gelatin) to 8% gelatin gel prepared in such a quantity to obtain 6% gelatin. This is described in details at the beginning of section 4.2
- In Table 2, please add the unit of measure. What are the differences with Table 1?
Table 1 shows the numerical values of the characteristics, while Table 2 shows the number of samples for each type of test. So in Table 2 unit of measure was quantity of samples in pieces .
Round 2
Reviewer 3 Report
Comments and Suggestions for Authors
The authors partially replied to the reviewer's comments, or other replies are unclear. More in details:
- In Table 1, are the measurements made on independent samples? Please also provide more details on the measurement setup. It appears strange that the control showed a higher standard deviation than the modified version of the control. Also, the number of samples to be compared among all groups must be similar. The reviewer suggests to improve the amount of analyzed samples, detail if these test were performed on independent samples, and report the type of measurement.
- The reply on the sample's rigidity is unclear. What do the authors mean by "those structures that are formed during its cross-linking" ?
- Is the lower presence of cells from day 7 up to 14 days is scientifically coherent with the intent of the application? Given the response provided by the authors, the reviewer suggests including a morphological analysis of all conditions to discern differences in the case studies analyzed.
- Figure 6 needs to be more detailed and clear.
- The analysis of the kinetics release of the antibiotics is essential, as well as the gelatin one, since the graft mechanical properties are dependent on that kinetics. The authors proposed an analysis at the time point 0 while to have a complete picture of the mechanical behaviour of the grafts over time, the knowledge of the mechanical properties after several days (e.g., 7 and 14 days) or the kinetic release of the components are determinant.
Author Response
- In Table 1, are the measurements made on independent samples? Please also provide more details on the measurement setup. It appears strange that the control showed a higher standard deviation than the modified version of the control. Also, the number of samples to be compared among all groups must be similar. The reviewer suggests to improve the amount of analyzed samples, detail if these test were performed on independent samples, and report the type of measurement.
All the measurement in Table 1 are made on independent samples. The detailed information about measurement methods and methods for preparing samples for experiments is contained in section 4 “Materials and methods”. Table 2 shows how many samples were used in each experiment. From this table 2 it is clear that the sample numbers compared among all groups were similar, excluding the control group when testing the kinking radius. There we increased the number of samples in order to most accurately determine the initial kinking radius. We do not understand the remark “It appears strange that the control showed a higher standard deviation than the modified version of the control.” Firstly, we did not have a “modified control”: there was a control and there were samples treated with different gelatins with/without different antibiotics. Secondly, it is not clear which indicator the reviewer is talking about. If he talks about permeability, then in group 6 sealed with GelB + Vancomycin the standard deviation (±6.1) is greater than in the control (±2.7). If he talks about the kinking radius, then in group 1 sealed with GelA the standard deviation (±4.0) is greater than in the control (±2.8). It is not clear for which tests the reviewer recommends improving the amount of samples.
- The reply on the sample's rigidity is unclear. What do the authors mean by "those structures that are formed during its cross-linking" ?
In the previous round of our answers, we provided a link to a work in which this question was most fully and clearly discussed. All sealing technologies using gelatin involve cross-linking the film formed on the prosthesis after impregnation with gelatin. Various bi- or polyfunctional chemical cross-linkers can be used for this: glutaraldehyde (GA), carbodiimide, epoxy compounds, etc. A physical method can also be used: dehydrothermal crosslinking. Various agents form different chemical structures in gelatin. Thus, glutaraldehyde reacts with the primary amino groups of lysine and hydroxylysine, carbodiimide cross-links at carboxyl groups, epoxy compounds react with primary and secondary amino groups and hydroxyl groups. Quality and quantity of cross-links determine the rigidity and hydrophilicity/hydrophobicity of the tissue. This depends on the crosslink density and the functional groups with which the cross-linker interacts. GA significantly increases the rigidity and hydrophobicity of both collagen and gelatin structures. This effect is well known. However, treatment technologies for prostheses intended for clinical use typically use glutaraldehyde. This linker is still the gold standard in the manufacturing of products for cardiovascular surgery, since it has been used for bioprosthetic heart valves preservation for almost half a century. In bioprostheses, GA cross-links collagen, which is the main fibrillar protein of the valves. Since gelatin is a collagen derivative and has a sufficient amount of free e-amino groups of lysine and hydroxylysine, its use for cross-linking a sealing gelatin film on vascular prostheses is quite acceptable. However, the side effect of increased rigidity always remains, despite the use of plasticizers (mainly glycerin, which we also used in our work).
- Is the lower presence of cells from day 7 up to 14 days is scientifically coherent with the intent of the application? Given the response provided by the authors, the reviewer suggests including a morphological analysis of all conditions to discern differences in the case studies analyzed.
Although the number of endothelial cells on woven vascular grafts with GelA or GelB, with or without antibiotics, decreases from days 7 to 14, the results obtained are scientifically coherent with the intent of the application. Sealing woven vascular grafts with GelA or GelB, with or without antibiotics, increases their cytocompatibility. On day 7, the number of adherent endothelial cells on all samples sealed with GelA or GelB, with or without antibiotics, was significantly higher compared to the control sample No. 0 (Figure 4, Table S3). The number of cells on all studied samples, whether sealed with GelA or GelB gelatin with or without antibiotics, decreases significantly by day 14 compared to day 7 (Fig. 4). However, it remains significantly higher compared to the control sample No. 0, except for samples Nos. 1 and 6.
It is not clear from the reviewer’s comment what exactly he means by morphological analysis, so we proceeded from the generally accepted concept that morphological analysis is a method for exploring possible solutions to a multi-dimensional, non-quantified complex problem using the technique of cross-consistency assessment. Of course, this is a very interesting approach. We appreciate the reviewer's suggestion to conduct a morphological analysis. We will consider this suggestion when planning further research on woven vascular grafts sealed with GelA or GelB, with or without antibiotics.
- Figure 6 needs to be more detailed and clear.
We tried to make Figure 6 more detailed and understandable for readers.
Figure 6. Fluorescence microscopy images of EA.hy926 cells on woven vascular grafts without gelatin and antibiotics (control sample No. 0); with GelA (samples Nos. 1-4) or GelB (samples Nos. 5-8) with or without antibiotics after three, seven, and fourteen days of culture. Staining was performed with acridine orange (green, live cells) and propidium iodide (yellow, apoptotic cells; red, dead cells). The scale bar is 50 µm.
- The analysis of the kinetics release of the antibiotics is essential, as well as the gelatin one, since the graft mechanical properties are dependent on that kinetics. The authors proposed an analysis at the time point 0 while to have a complete picture of the mechanical behaviour of the grafts over time, the knowledge of the mechanical properties after several days (e.g., 7 and 14 days) or the kinetic release of the components are determinant.
With all due respect, the kinetics of antibiotic release is in no way capable of influencing the mechanical properties of prostheses, because these antibiotics are completely water-soluble substances, and the prosthetic mechanics has nothing to do with their presence or absence in the swollen gelatin gel. The dry gelatin film, formed as a result of sealing followed by drying, when exposed to a water-containing medium (blood, water, culture medium, etc.) swells very quickly (minutes) and turns into a gel. This swelling radically changes the mechanical properties of the woven prosthesis, such as rigidity. The prosthesis immediately becomes soft and its kinking radius decreases. At the same time, the gelatin gel maintains the low permeability of the prosthesis, since cross-linked gelatin is able to close the pores until it is completely desorbed (which takes several days, according to the literature - about 8 days). These are well-known facts and do not need to be verified.
As for the kinking radius, it must be measured dry in accordance with the ISO 7198 (i.e., “at the 0 point”). The reviewer suggests assessing the mechanical behavior of the prosthesis after 7-14 days. In what medium should the prosthesis be placed: in water, in blood under static incubation, in the blood stream in vitro, in the blood stream after orthotopic implantation? Results strongly depend on the medium and functioning (load, no load), but will vary significantly. The most justified would be functioning in the conditions for which the prosthesis is intended - i.e., for example, after 7 and 14 days of implantation into the thoracic aorta of large animals or humans. However, in this case, the results will be affected by many factors: first of all, intensive wall impregnation with blood proteins and cells. In this regard, we are unlikely to learn anything new about the effect of gelatin and antibiotics on the mechanical behavior of the prosthesis over time.
Round 3
Reviewer 3 Report
Comments and Suggestions for Authors
The authors did not clarify all the reviewer's requests of the previous round of revision. Below the authors can find some further considerations:
- With the morphological analysis of cells, the reviewer intended the interpretation of the different cell adhesion/morphology when seeded onto the substrates under analysis.
- Regarding your sentence: "the kinetics of antibiotic release is in no way capable of influencing the mechanical properties of prostheses, because these antibiotics are completely water-soluble substances". The antibiotic concentration is quite relevant with respect to the other components, and the authors showed their influence on the mechanics of the prosthesis, and how the kinking radius was altered from gelatin A and B alone. Above all, the rigidity you considered (but not reported in terms of Young's modulus) is in a dry state, while in the real application, there will be softening due to the swelling of gelatin and the release of antibiotics, as stated by the authors. Will the stiffness of all cases remain the same after antibiotics dissolution?
- For the kinking analysis, the medium to be used should be dictated by the potential application proposed by the authors, not by the reviewer.
Comments on the Quality of English Languagenothing to add
Author Response
The authors did not clarify all the reviewer's requests of the previous round of revision. Below the authors can find some further considerations:
- With the morphological analysis of cells, the reviewer intended the interpretation of the different cell adhesion/morphology when seeded onto the substrates under analysis.
The authors demonstrated different endothelial cell adhesion on tissue vascular grafts with GelA or GelB, with or without antibiotics, by counting the number of adherent cells (Fig. 4). Cell adhesion was also confirmed using fluorescence microscopy (Fig. 6). The morphological analysis proposed by the reviewer is an additional method for confirming differences in adherent cells. We usually use DAPI stain for this and always note differences if they are present (Zhuravleva IY, Surovtseva MA, Vaver AA, Suprun EA, Kim II, Bondarenko NA, Kuzmin OS, Mayorov AP, Poveshchenko OV. Effect of the Nanorough Surface of TiO2 Thin Films on the Compatibility with Endothelial Cells. Int J Mol Sci. 2023 Apr 3;24(7):6699. doi: 10.3390/ijms24076699). However, in this study there were no such differences. We did not include this series of experiments in the article because we did not want to overload it with irrelevant information.
- Regarding your sentence: "the kinetics of antibiotic release is in no way capable of influencing the mechanical properties of prostheses, because these antibiotics are completely water-soluble substances". The antibiotic concentration is quite relevant with respect to the other components, and the authors showed their influence on the mechanics of the prosthesis, and how the kinking radius was altered from gelatin A and B alone. Above all, the rigidity you considered (but not reported in terms of Young's modulus) is in a dry state, while in the real application, there will be softening due to the swelling of gelatin and the release of antibiotics, as stated by the authors. Will the stiffness of all cases remain the same after antibiotics dissolution?
Apparently, the reviewer was misled by the term “stiffness” (Line 312), which we certainly used incorrectly. We are talking about kink resistance; this property is related to stiffness only indirectly and is tested without using Young’s modulus. “Stiffness is the extent to which an object resists deformation in response to an applied force. In other words stiffness is a measure of how much force it takes to deform a body by a given amount. Rigidity is the property of a structure or a body that it does not bend or flex under an applied”. We have corrected this gap in the text. As for testing in the dry state, it is here that the differences in kink resistance of woven synthetic prostheses are most fully manifested. In a wet state they are leveled out, which is why ISO 7198 prescribes testing such prostheses in a dry state.
- For the kinking analysis, the medium to be used should be dictated by the potential application proposed by the authors, not by the reviewer.
When testing vascular grafts, it is correctly to use ISO 7198, and the authors of most articles do so (e.g. Valsecchi, E.; Biagiotti, M.; Alessandrino, A.; Gastaldi, D.; Vena, P.; Freddi, G. Silk Vascular Grafts with Optimized Mechanical Properties for the Repair and Regeneration of Small Caliber Blood Vessels. Materials 2022, 15, 3735. https://doi.org/10.3390/ma15103735. Lucereau B, Koffhi F, Lejay A, Georg Y, Durand B, Thaveau F, Heim F, Chakfe N. Compliance of Textile Vascular Prostheses Is a Fleeting Reality. Eur J Vasc Endovasc Surg. 2020;60(5):773-779. doi: 10.1016/j.ejvs.2020.07.016). This is necessary that the results obtained by different scientific groups can be compared. According to ISO 7198:2016 (Annex A, Subclause A.5.8.), synthetic woven and knitted vascular grafts must be tested for kink resistance in a dry state. Thus, the medium to be used for testing are dictated not by the authors, not by the reviewer, but by the international standard.